# Active Low-Density Polyethylene-Based Films by Incorporating α-Tocopherol in the Free State and Loaded in PLA Nanoparticles: A Comparative Study

**DOI:** 10.3390/foods13030475

**Published:** 2024-02-02

**Authors:** Ana G. Azevedo, Carolina Barros, Sónia Miranda, Ana V. Machado, Olga S. Carneiro, Bruno Silva, Mariana A. Andrade, Fernanda Vilarinho, Margarida Saraiva, Ana Sanches Silva, Lorenzo M. Pastrana, Miguel A. Cerqueira

**Affiliations:** 1International Iberian Nanotechnology Laboratory, Av. Mestre José Veiga s/n, 4715-330 Braga, Portugal; ana.azevedo@inl.int (A.G.A.); lorenzo.pastrana@inl.int (L.M.P.); 2Institute for Polymers and Composites, University of Minho, Campus de Azurém, 4800-058 Guimarães, Portugal; carolina.barros@piep.pt (C.B.); avm@dep.uminho.pt (A.V.M.);; 3Centre for Innovation in Polymer Engineering, University of Minho, Campus de Azurém, Edifício 15, 4800-058 Guimarães, Portugal; sonia.miranda@piep.pt (S.M.); bruno.silva@piep.pt (B.S.); 4National Institute of Health Doutor Ricardo Jorge, Avenida Padre Cruz, 1649-016 Lisboa, Portugal; mariana.andrade@insa.min-saude.pt (M.A.A.); fernanda.vilarinho@insa.min-saude.pt (F.V.); 5Associated Laboratory for Green Chemistry of the Network of Chemistry andTechnology (REQUIMTE/LAQV), R. D. Manuel II, Apartado, 55142 Porto, Portugal; 6National Institute of Health Doutor Ricardo Jorge, Rua Alexandre Herculano, 321, 4000-055 Porto, Portugal; 7Pharmacy Faculty, University of Coimbra, Polo III, Azinhaga de Stª Comba, 3000-548 Coimbra, Portugal; asanchessilva@ff.uc.pt; 8Center for Study in Animal Science (CECA), ICETA, University of Porto, 4501-401 Porto, Portugal; 9Associate Laboratory for Animal and Veterinary Sciences (AL4AnimalS), 1300-477 Lisbon, Portugal

**Keywords:** α-tocopherol, biomaterial, active packaging, nanomaterial

## Abstract

In this work, alpha-tocopherol (α-TOC) was encapsulated in poly(lactic acid) nanoparticles (PLA NPs) and added to low-density polyethylene (LDPE) films with the aim of producing an active film for food packaging applications. PLA NPs loaded with α-TOC were produced through nanoprecipitation and dried using two methods (freeze-dryer and oven). LDPE-based films with final polymeric matrix concentrations of 10 and 20 g/kg were then produced through blow extrusion. The results showed that LDPE-based films loaded with α-TOC can be produced using blow extrusion, and a good distribution of PLA NPs can be obtained within the LDPE matrix as observed using scanning electron microscopy (SEM). The mechanical properties were affected by the incorporation of α-TOC and PLA NPs loaded with α-TOC, with the observation of a decrease in tensile strength and Young’s Modulus values and an increase in elongation at break. Regarding water vapor permeability, the films showed a reduction in the values with the addition of α-TOC and PLA NPs loaded with α-TOC compared to the LDPE film (control). Films with α-TOC in the free state and loaded in PLA NPs showed antioxidant activity, but their behavior was affected by the encapsulation process.

## 1. Introduction

The use of active substances in packaging materials has been pointed out as an attractive method to produce new food packaging materials with the capacity to extend food shelf-life [1]. Particularly, antioxidants have been studied to prevent the negative impact of oxidation reactions on food quality during food storage. Antioxidant agents are added directly to food or food packaging to inhibit or slow down the oxidation reactions. They react with reactive oxidizing species (e.g., peroxides, superoxides, and hydroxyl radicals), slowing or blocking the oxidation reactions of food products. When the antioxidant agents are added to the packaging materials, they are released from the packaging material to the headspace, by vaporization, or diffuse or migrate into the food [2].

In recent years, there has been an increasing interest in replacing synthetic antioxidants, such as butylated hydroxytoluene (BHT), butylated hydroxyanisole (BHA), and tert-butylhydroquinone (TBHQ), with natural antioxidants, like a plant extract and essential oils. One of the most studied natural antioxidant compounds in the food area is tocopherol (or Vitamin E), which has some applications in active food packaging [3]. Tocopherol includes different natural forms, such as α-, β-, γ-, and δ-tocopherol, and it can be found in several products with fats, such as in olive or almond oil, hazelnuts, and egg yolk. Among all the forms, alpha-tocopherol (α-TOC) is the most common and biologically active form. α-TOC is a phenolic antioxidant and its active mechanism involves the donation of hydrogen from the hydroxyl group (-OH) of the phenolic ring to free radicals (ROS) [4]. Several studies have shown a good antioxidant activity using the α-TOC [5,6,7,8].

In addition to replacing the synthetic antioxidants, there is also the concern about replacing the petroleum-based polymers used to produce food packaging with bio-based polymers, such as polylactide acid (PLA) and polyhydroxyalkanoates (PHAs) [8,9,10,11]. However, these solutions are still not common, because they are expensive when compared with the synthetic alternatives. For this reason, synthetic polymers, such as low-density polyethylene (LDPE) and high-density polyethylene (HDPE), are still extensively used in film packaging [12,13]. However, to reduce the negative impact of synthetic polymers, the scientific community and companies have tried to find some strategies, like reducing the number of layers in multilayer films and reducing the use of different materials, towards using monomaterials that are easily recyclable or/and compostable. One strategy to solve this is the addition of fillers or nanomaterials that can guarantee the good properties of multilayer films. Another approach is the use of nanoparticles or carriers loaded with active agents. In the literature, studies that incorporate active agents into carriers or nanoparticles have focused on absorption in porous media materials [14,15,16] and integration through ultrathin fibers [10,17,18]. More rarely, the encapsulation of active compounds in hollow polymeric materials bodies, namely small capsules or nanoparticles, has been studied. The production of capsules or particles can be performed using different methodologies, such as the ionic gelation technique and emulsification [19,20].

In the current study, the nanoprecipitation process, also known as the solvent displacement method, described first by Fessi et al. [21], was used. This consists of a method for the development of nanoparticles in a reproducible, easy, and scalable way. The nanoprecipitation method requires three basic ingredients: the polymer, the solvent, and the antisolvent (also known as non-solvent) of the polymer. A suitable solvent should be completely miscible with the antisolvent, and able to solubilize the bioactive compound. Moreover, the antisolvent must have a low boiling point, allowing its removal by evaporation. The polymer, the bioactive compound to be encapsulated, and the solvent constitute the organic phase. The antisolvent of the polymer constitutes the aqueous phase, which is usually water. Several operating conditions influence the characteristics of the nanoparticles, such as the ratio of the organic phase to the aqueous phase, the stirring rate, the injection rate of the organic phase (which is poured), and the polymer concentration [22].

One of the challenges of producing composites and nanocomposites is the use of scalable technologies, such as extrusion processes, where the mixed compounds are processed by extrusion and used to produce films and sheets. In this regard, only a few works have explored the use of extrusion to produce active films with tocopherol-loaded micro- and nanostructures. For example, Gargiulo et al. [14] produced mesoporous silica loaded with α-TOC and added it to an LDPE matrix using an internal mixer. The compounding obtained was then fed into a co-rotating laboratory twin-screw extruder equipped with a sheet die, and active antioxidant films were produced. Sun et al. [15,23,24,25] reported a similar study using LDPE films with α-TOC impregnated into mesoporous silica. First, the α-TOC that was adsorbed on mesoporous silica was mixed with LDPE pellets using twin-screw extrusion, and then the active film was produced using compounding through flat extrusion. The studies showed a slower antioxidant release of α-TOC that was loaded into a silica substrate compared to the samples containing free tocopherol. However, it was also demonstrated that the release of α-TOC depends on the size of the mesoporous silica.

In the case of Li et al. [26], they studied an antioxidant LDPE film containing α-TOC and quercetin loaded in mesoporous silica. The mesoporous silica loaded with the active agents was added directly into LDPE using a twin-screw extruder and then the compounding was processed in the cast extruder to produce the LDPE films. The release results showed that α-TOC loaded in mesoporous silica with quercetin was reduced compared with α-TOC and quercetin without being loaded in mesoporous silica. In addition, the oxygen barrier was improved with α-TOC loaded in mesoporous silica with quercetin.

In this work, we aimed to understand how tocopherol-loaded nanoparticles could influence the physicochemical and active properties of LDPE-based films. For this purpose, PLA-based nanoparticles were produced by the nanoprecipitation process and loaded with α-TOC. PLA NPs loaded with α-TOC and in the free state were then incorporated into the LDPE matrix using an extrusion process. The effect of α-TOC encapsulated in PLA NPs and in the free state was evaluated regarding the films’ optical, mechanical, and barrier properties. In addition, the active properties of the films were evaluated using the β-carotene bleaching assay and by the determination of total phenolic compounds.

## 2. Materials and Methods

### 2.1. Materials

An amorphous poly(lactic acid) (PLA) with a high viscosity, density of 1.24 g/cm^3^, and molecular weight of 1.63 × 10^5^ kDa was obtained from Total Corbion company (Gorinchem, The Netherlands). Alpha-tocopherol (α-TOC) was purchased from BTC Europe GmbH (Monheim, Germany), low-density polyethylene (LDPE) was obtained from Exxon Mobil Corporation (Houston, TX, USA), polyethylene modified with maleic anhydride (PE-g-MA compatibilizer, FUSABOND) was obtained from Dow company (Hayward, CA USA), acetone was obtained from Honeywell (Charlotte, NC, USA), and ultra-pure water was obtained through a Milli-Q^®^ purification system (Millipore Corp., Belford, NJ, USA).

### 2.2. PLA Nanoparticles’ Production Loaded with α-TOC by Nanoprecipitation

The production of the PLA NPs followed the methodology described by Fessi et al. [21] and Afonso et al. [22]. Before loading the PLA NPs with α-TOC, different approaches were followed to find the parameters that allow a high yield of production). To produce PLA NPs, PLA pellets were dissolved at 0.5% (*w*/*v*) in acetone under stirring (300 rpm) at 70 °C in a water bath. After PLA dissolution, the solution was cooled down, and α-TOC was added and stirred for 30 min. Afterwards, 20 mL of this PLA solution was added to 80 mL of ultrapure water, used as an antisolvent, using an Ultra-turrax (Ultra-Turrax T18 digital with S10G dispersing element, IKA^®^-Werke GmbH & Co. KG (Königswinter, Germany) at 11,000 rpm, for producing PLA NPs loaded with α-TOC. After this production process, the acetone was evaporated using a rotary evaporator (IKA^®^-Werke GmbH & CO. KG, Germany) at 60 °C, until the acetone was removed. After the evaporation process, the resulting solutions of PLA NPs loaded with α-TOC were dried using two alternative processes: freeze-drying and oven-drying. In the first case, the loaded PLA NP solution was frozen at −20 °C and then put in freeze-dryer Lyoquest −55 °C Plus Eco equipment (Telstar, Terrassa, Spain) where the PLA NPs were dried through a sublimation process. In the second case, the solution of loaded PLA NPs was dried through the evaporation process in an oven at 60 °C. The production yield was calculated according to Equation (1):(1)$$Production\;yield\;(\%)=\frac{Weigth\;of\;PLA\;NPs\;obtained}{Weight\;of\;PLA\;initial}\times 100$$

### 2.3. Characterization of PLA NPs

#### 2.3.1. Dynamic Light Scattering

Dynamic light scattering (DLS) (SZ-100Z, Horiba Instruments Inc., Kyoto, Japan) was used for the determination of nanoparticle size and polydispersity index (PDI). All measurements were performed at a temperature of 25 °C, actively maintained within 0.1 °C in the sample chamber using polystyrene cuvettes with four openings. The samples were irradiated with a diode-pumped frequency-doubled laser (532 nm, 10 mW) and the intensity fluctuations of the scattered light were detected at an angle of 173°. The particle refractive index used was 1.48 for PLA and 1.33 for the dispersion medium water. Software (Horiba SZ-100Z Type) was used to determine the mean size according to the diffusion coefficient, using the Stokes–Einstein equation. The measurements were performed immediately after the production process without diluting the samples. No sedimentation was observed during the measurements. For each sample, at least five measurements were performed.

#### 2.3.2. Electron Microscopy

Transmission electron microscopy

The morphology of the unloaded and loaded PLA NPs was evaluated using transmission electron microscopy (TEM) (JEOL JEM 2100-HT—200 kV LaB6 gun, JEOL Ltd., Tokyo, Japan). The samples were deposited onto grids coated with an ultrathin carbon film (400 mesh, approx. grid hole size of 42 μm, PELCO^®^, Ted Pella Inc., Redding, CA, USA) and UranyLess EM Stain (Electron Microscopy Sciences, Hatfield, PA, USA) was used as the contrast agent. The samples were dried at room temperature and the micrographs of the samples were taken after 24 h. The images were digitally recorded using an UltraScan^®^ 4000 CCD camera (Oneview, Gatan, USA).

Scanning electron microscopy

The morphology of PLA NPs loaded with α-TOC in powder was evaluated using scanning electron microscopy (SEM) (QUANTA 650FEG—FEI Europe B.V. Company (Eindhoven, The Netherlands). The PLA NPs were coated with 10 nm gold (Au) and examined on the surface.

#### 2.3.3. Encapsulation Efficiency and Loading Capacity

Encapsulation efficiency (EE) and loading capacity (LC) were determined according to Azevedo et al. [27], using Amicon^®^ Ultra—0.5 mL 100 kDa device (Millipore Corp., Cork, Ireland). Briefly, 0.5 mL of loaded PLA NP solution was added to the Amicon^®^ and centrifuged at 14,000× *g* during 15 min. From centrifugation, a filtrate with free α-TOC and a concentrate with PLA-NP-encapsulated α-TOC were obtained. The filtrate with free α-TOC was used to calculate the EE. For this reason, it was analyzed using spectrophotometry using an UV—Visible microtiter plate reader (Synergy H1, BioTek Instruments, Winooski, VT, USA) at 285 nm (maximum absorbance peak determined). The amount of free α-TOC in the filtrate was calculated through a calibration curve y = 9.0285x – 0.0092 (R^2^ =0.999). The concentrate with PLA-NP-encapsulated α-TOC was dried and used to determine the LC. EE and LC were calculated using Equations (2) and (3), respectively:(2)$$EE\;(\%)=\frac{TOC\;initial-TOC\;free}{TOC\;initial}\times 100$$
(3)$$LC\;(\%)=\frac{TOC\;initial-TOC\;free}{NPs\;total}\times 100$$
where α-TOC initial represents the total amount of α-TOC initially added, α-TOC free represents the amount of free α-TOC in the filtrate, and NPs total corresponds to the weight of dried nanoparticles.

#### 2.3.4. Thermogravimetric Analysis (TGA)

The thermal properties were measured using thermal gravimetric analysis (TGA) using a TGA Q500 (TA Instruments, New Castle, DE, USA). The measurements were carried out from 30 to 600 °C, with an increasing rate of 10 °C/min, under nitrogen atmosphere.

### 2.4. Incorporation of PLA NPs Loaded with α-TOC into LDPE Matrix and Film Production

The PLA NPs loaded with α-TOC, dried by freeze-drying and oven-drying, were incorporated into the LDPE matrix using a twin-screw extruder (Leistritz GA, Nuremberg, Germany) with a temperature profile of 140–180 °C. The incorporation rates of loaded PLA NPs were 10 and 20 g/kg of polymer matrix, which corresponds to 0.01 and 0.02 g of active agent per g of material. After, the different compounds obtained were used to produce active films using a blown film extrusion line (from Periplast, Leiria, Portugal) with a temperature profile of 170–180 °C. LDPE without PLA NPs was used as control, and LDPE film with α-TOC in the free state was produced using 10 and 20 g/kg polymer matrix, in order to compare its behavior in the free state and when encapsulated. In all the samples, 5% PE-g-MA of compatibilizer was added. This value was selected based on the literature that reported good dispersion properties for this concentration [28,29].

Descriptions and notations of the films are given in Table 1.

### 2.5. Film Characterization

#### 2.5.1. Optical Properties

All the films were analyzed visually and some pictures were taken to compare with the LDPE control film.

The haze was determined according to ASTM D1003-13 [30] in an XL-211 Hazegard System transmittance meter (BYK Instruments, Columbia, MD, USA). This system measures the total light transmittance (Tt) and the percentage of diffuse transmittance, Td. Six specimens from each film were tested. Percent haze was calculated using Equation (4):(4)$$Haze=\frac{Td}{Tt}\times 100$$

Color measurements were performed in a Minolta colorimeter (Minolta CR 400, Tokyo, Japan). The films were superposed on a white and black standard, which recorded the spectrum of reflected light and converted it into a set of color coordinates (L*, a*, and b* values). Three measurements were taken of each sample. The total difference in color (ΔE) was determined using Equation (5):(5)$$\Delta E=\sqrt{{\left(∆L\right)}^{2}+{\left(∆a\right)}^{2}+{\left(∆b\right)}^{2}}$$

#### 2.5.2. Scanning Electron Microscopy

The morphology of LDPE films with α-TOC in the free state and PLA NPs loaded with α-TOC was analyzed, in the cross-section, using scanning electron microscopy (SEM) (QUANTA 650FEG—FEI Europe B.V. Company, Eindhoven, The Netherlands). The cross-sections were prepared using liquid nitrogen to freeze and fracture the film. After, all the films samples were coated with 10 nm of gold.

#### 2.5.3. Mechanical Properties

The mechanical properties were determined based on the ISO 527-1/-3 standard [31,32], using a Shimadzu AG-X universal mechanical testing machine (Shimadzu Corporation, Kyoto, Japan), with a 1 kN load cell and pneumatic tethers. The tests were carried out at a speed of 50 mm/min and a temperature of 23 °C. To determine the secant modulus at 1% strain, a Shimadzu SIE-560 contact strain gauge with an initial distance of 50 mm was used. Ten specimens of type 2 of the ISO 527-3 [32] standard were tested in the longitudinal (LD) and transversal (TD) directions.

#### 2.5.4. Water Vapor Permeability

Water vapor permeability was gravimetrically determined according to the ASTM E96/E96M [33], using the water method. Films were sealed on test dishes (Elcometer 5100 Payne Permeability Cups) with distilled water, and the assembled test dishes were placed into a desiccator, at 21 (±1) °C and 0 (±2) % relative humidity (RH), with silica gel. The assembled test dishes were weighed once a day and weight loss was measured over time until steady state was reached. Water vapor permeability (WVP) (g/(m.s.Pa)) was determined using Equation (6):(6)$$WVP=\frac{\left(WVT\times L\right)}{\Delta P}$$
where WVT is the water vapor transmission rate ((g/(h m^2^)), L is the film thickness (m), and ΔP is the water vapor partial pressure difference (Pa) across the two sides of the film.

#### 2.5.5. Antioxidant Activity

β-carotene bleaching assay

For the β-carotene bleaching assay, the method described by Andrade et al. [34] was applied. First, β-carotene diluted in chloroform solution (2 mg/mL) was prepared. Then, to form an emulsion, 1 mL of the β-carotene solution, 20 mg of linoleic acid, and 200 mg of Tween^®^40 were mixed. The chloroform was evaporated in a rotary evaporator at 35 °C and 50 mL of ultrapure water were mixed and vigorously shaken. Then, 200 μL of sample (for the control, 200 μL of ethanol 95% was used), 5 mL of emulsion was added, and the mixtures were exposed to 50 °C for 2 h. The absorbance was then measured at 470 nm at the end of this time. The absorbance of the controls was measured before and after the heating period. The antioxidant activity coefficient (AAC) was calculated through Equation (7):(7)$$AAC=\frac{Aa-Ac2}{Ac0-Ac2}\times 1000$$
where Aa stands for the absorbance of the sample, Ac0 stands for the absorbance of the control before the heating period, and Ac2 stands for the control absorbance at the end of the heating period.

Total phenolic compounds

For the quantification of the total phenolic compounds (TPCs), the methods described by Erkan et al. [35] were performed. An amount of 1 mL of sample was mixed with 7.5 mL of a Folin–Cioucalteu aqueous solution (10%, *v*/*v*) and homogenized, and, after 5 min, 7.5 mL of an aqueous solution of sodium carbonate (60 mg/mL) was added. The solutions were homogenized and left to stand at room temperature, protected from the light, for 2 h. At the end of this period, the absorbance was measured at 725 nm. A calibration curve, using gallic acid as standard, was drawn. The results are presented in gallic acid equivalents per mL (μg/mL).

### 2.6. Statistical Analysis

The statistical analysis was performed using one-way analysis of variance (ANOVA), and Tukey’s multiple comparison test was performed to determine the significant differences (*p* < 0.05) between film samples.

## 3. Results

### 3.1. Characterization of Unloaded and Loaded PLA Nanoparticles

PLA NPs loaded with α-TOC were produced to incorporate into LDPE films. The loaded PLA NPs were dried using an oven and freeze-dryer. The visual appearance of the loaded PLA NPs was different, depending on the drying process used. The PLA NPs dried using an oven had a yellow color, while the PLA NPs dried using the freeze-dryer had a white color (Appendix A). Then, different films using both types of dried loaded PLA NPs were produced. The loaded PLA NPs were first mixed with LDPE polymer, after which active films were produced. All the films were characterized in terms of optical, mechanical and barrier properties and antioxidant activity.

Preliminary results showed that 0.5% (*w*/*v*) of PLA, 0.5% (*w*/*v*) of α-TOC, and a volume ratio of 20/80 (S/AS—solvent/antisolvent) allowed us to obtain a high production yield (Table 2). Moreover, these conditions allowed us to produce loaded PLA NPs with an average size ranging between 100 and 200 nm. However, it can be seen that the addition of α-TOC affected the size of PLA NPs, which increased from 122 nm to 187 nm. The values of the PDI and production yield did not show significant differences. The results reported by Maharana et al. [36] showed similar values of the NP size, PDI, and production yield (111 nm, 0.1, and 90%, respectively) when similar conditions were used (PLA concentration of 5 mg/mL, acetone solvent, and S/AS volume ratio of 0.20). However, those authors used a lower molecular weight of PLA (98.470 kDa) compared to that used in the current study. This can explain the difference observed in the values, because they demonstrated that the molecular weight of PLA can affect the size of the PLA NPs and yield.

The morphology of unloaded and loaded PLA NPs was also analyzed using TEM. Figure 1a,b show TEM images that confirm the spherical shape for both cases. The morphology observed is in agreement with the PLA NP TEM images presented elsewhere [36]. The images also showed a broad range of particle sizes, in accordance with the polydispersity values obtained using DLS. The NP sizes obtained seem to be lower than the DLS values, around 100 and 140 nm for unloaded PLA NPs and loaded PLA NPs, respectively. This can be explained by the drying step needed for the sample preparation for TEM analysis.

PLA NPs loaded with α-TOC were also analyzed using SEM (Figure 2). Figure 2a,b show the PLA NPs dried using an oven and freeze-dryer, respectively, and in both cases, agglomeration of the nanoparticles is exhibited. Although the particles appear to have a uniform and spherical shape, their measurement was not possible, due to agglomeration. These images are similar to those found by Lasalle and Ferreira [37] who used the nanoprecipitation method to produce PLA nanoparticles. They also observed the agglomeration of the particles, but found particles to be uniform and spherical, with sizes smaller than 200 nm.

The thermal behavior of the α-TOC, unloaded PLA NPs, and PLA NPs loaded with α-TOC is presented in Figure 3. TG curves of the α-TOC and unloaded PLA NPs showed a single thermal event related to the degradation of α-TOC and PLA NPs, respectively, at which the initial degradation temperature was near 220 °C. Concerning the loaded PLA NPs, dried using an oven and freeze-dryer, they also showed a single decomposition stage corresponding to the degradation of the α-TOC and PLA NPs. However, the encapsulation process affected the initial degradation temperature that decreased to around 200 °C. So, the result showed that the encapsulated process was not beneficial for α-TOC stability, as demonstrated by Sun et al. [15] and Li et al. [26] where they showed that α-TOC loaded onto mesoporous silicate (MCM) was more stable at higher temperature. This can be explained by changes in the PLA structure, namely crystallinity, with the addition of α-TOC [8,38]. However, this thermal analysis showed, therefore, that both materials, α-TOC and PLA, and α-TOC encapsulated in PLA NPs could be able to resist the typical temperatures used in the extrusion process of LD-PE.

The ability of PLA NPs to encapsulate α-TOC was evaluated through the determination of encapsulation efficiency (EE) and loading capacity (LC). The obtained values of EE and LC were 98.80 ± 0.07% and 49.79 ± 10.63%, respectively. The high value of EE is in line with the literature (higher than 90%) and can be explained by the lipophilic properties of α-TOC that help to obtain a good EE [36,37]. Regarding the value of LC, a value lower than 20% was found elsewhere using the nanoprecipitation method, while in the current study, a value around 50% was obtained. This can be explained by the good compatibility between PLA and α-TOC [39].

### 3.2. Characterization of LDPE Films

Figure 4 presents the various types of LDPE films produced over a background image. In general, the pictures show that the transparency of the films was not affected by the addition of the free-state α-TOC and loaded PLA NPs, when compared with the LD PE control film. However, the addition of α-TOC affected the turbidity of films (*p* < 0.05), as presented in Table 3, which increases for higher concentrations of α-TOC.

The color of the films containing α-TOC was slightly yellowish. This modification was only detectable when films were rolled up. The values of the color difference (ΔE), compared to the control film, are shown in Table 3. The values showed a significant difference (*p* < 0.05) when α-TOC was added in the free state and the PLA NPs. This result is in line with the studies performed by Wessling et al. [40] and Manzanarez-Lopez et al. [8]. In the first case, the authors showed that the addition of α-TOC into LDPE film promoted a change in (ΔE) of around 0.1–0.4, depending on the α-TOC concentration. In the second case, it was shown that it promoted the yellowness (b*) of the film, causing a color difference (ΔE) of 0.54 when compared to the PLA control film.

Figure 5 shows SEM images of the different films under study. The LDPE film with free-state α-TOC does not show differences between the matrix and α-TOC. Therefore, it is possible to confirm that the incorporation of α-TOC into the matrix was good. Concerning the images of LDPE films with loaded PLA NPs, the NPs showed a good distribution in all the films. The images showed PLA NPs with an average size of 300 (n = 5). Despite the values being higher than those determined by DLS after the production of NPs, it can be expected that different sizes are observed (due to the size distribution), but also some agglomeration of NPs occurs during the drying and extrusion process.

Table 4 presents the tensile strength (TS), Young’s Modulus (YM), and elongation at break (EB) of the films. The TS values in the longitudinal direction (LD) decreased and that in the transversal direction (TD) increased when α-TOC in the free state and PLA NPs were added to the films. These results showed a significant difference (*p* < 0.05) between the LDPE control film and all other films in LD and TD. Concerning the values of YM, in general, the same situation occurred as TS, since the values decreased in LD and increased in TD. Only in the TD direction did the results show no significant difference (*p* > 0.05) between the values obtained from the control film and the film with α-TOC in the free state (20_Free) and 10_PLA_O. In the case of EB, the values increased with the incorporation of α-TOC in the free state and PLA NPs for some of the films in LD and TD. Curiously, these increases took place with low concentrations of α-TOC and PLA NPs and showed a significant difference (*p* < 0.05) when compared to the LDPE control film. These results are in agreement with the study of Wessling et al. [40] which studied films containing α-TOC in LD and TD. Briefly, they showed a slight increase in the TS of the films containing α-TOC in TD and a decrease in YM in the LD and TD. Regarding the EB results, they showed a slight increase for films with α-TOC in the free state in TD.

Figure 6 shows the results of the water vapor permeability (WVP) of LDPE films. In general, the WVP values decreased slightly with the addition of free and encapsulated α-TOC when compared to the LDPE control film. This maximum difference occurred in loaded PLA NPs films dried using the freeze-dryer. These results agree with those published by Sun et al. [15] who showed a decrease in WVP, from 4.91 to 4.85 × 10^−5^ g.cm/cm^2^.s.Pa in LDPE films with α-TOC in the free state. However, the same authors observed a WVP increase, from 4.91 to 10.39 × 10^−5^ g.cm/cm^2^.s.Pa, in LDPE films with α-TOC incorporated in mesoporous silica. They explained that this increase could be partly due to the hydrophilic nature of mesoporous silica caused by abundant silanol groups on the surface of its pore wall. In the current study, the decrease in WVP with PLA NPs can be explained by the good distribution of the PLA NPs that create a tortuous path for the water molecules and the lipophilic behavior of α-TOC.

### 3.3. Antioxidant Activity of LDPE Films

The results of the antioxidant capacity of the active LDPE films with free-state α-TOC and when encapsulated are shown in Table 5. Preliminary results (results not shown) confirmed that LDPE did not present any antioxidant activity, as reported elsewhere [15]. These were analyzed through the β-carotene assay and total phenolic compounds (TPCs) for 3 and 10 days at 40 °C, in contact with ethanol 95 % (*v*/*v*). In the β-carotene bleaching assay, the ability of antioxidant compounds to inhibit lipid peroxidation was assessed. In the first 3 days, the loaded PLA NPs films showed a higher antioxidant capacity than the free-state α-TOC films. However, the difference is only significant (*p* < 0.05) between films with 10 free-state α-TOCs and 10 α-TOCs encapsulated in PLA NPs, dried using an oven. For the remaining samples, no significant differences were observed (*p* > 0.05).

After 10 days, all the samples showed increased antioxidant activity, but the films with free-state α-TOC showed remaining activity. These may be explained by the immediate availability of the α-TOC, when it was in the free state, compared to the slow release when it was encapsulated in PLA NPs. The slow release can be an advantage for food packaging since it can guarantee an active material for a greater length of time. A similar behavior was reported by Sun et al. [15] when they studied the LDPE with α-TOC adsorbed on mesoporous silica. They reported that the α-TOC adsorbed on mesoporous silica was released slower than in the free state.

Concerning the TPC after 3 days, the highest content was obtained in films with free-state α-TOC with significant differences (*p* < 0.05). However, after 10 days, the encapsulated formulations presented the highest values of the TPC. The film with the higher concentration of the PLA NPs dried using the freeze-dryer showed the best value, but the statistical analysis showed only a significant difference between this sample and the film with 10 free-state α-TOCs (*p* < 0.05).

These results show that the α-TOC is more stable during long periods when encapsulated, therefore being adequate for increasing product shelf life in food packaging.

## 4. Conclusions

Active LDPE was successfully produced through extrusion using TOC as the active agent. TOC was incorporated in the free state and loaded in PLA NPs, with the PLA NPs being homogenously distributed in the LDPE matrix. The mechanical results showed that the incorporation of α-TOC in the free state or when encapsulated in PLA NPs affects the film’s properties to some extent. While a decrease in the values for the EB for TS and YM was observed, it seems that the encapsulation of α-TOC in the PLA NPs decreases the plasticizer effect of the α-TOC. In addition, the LDPE active films obtained showed antioxidant properties and improved water vapor barrier properties.

This work opens new possibilities for using biodegradable polymers as carriers of active compounds to develop active food packaging. While the advantages of using the NPs are not clear in LDPE films, it would be important to test other materials and evaluate oxygen permeability and the migration of α-TOC to a food simulant.

## Figures and Tables

**Figure 1 foods-13-00475-f001:**
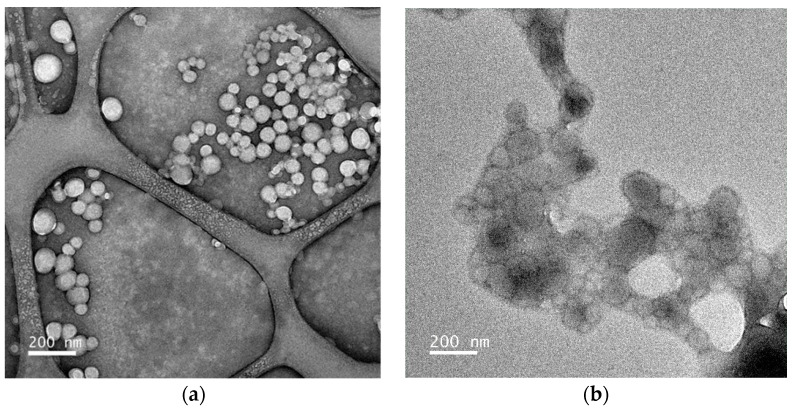
TEM image of (**a**) unloaded PLA NPs and (**b**) loaded PLA NPs with α-TOC.

**Figure 2 foods-13-00475-f002:**
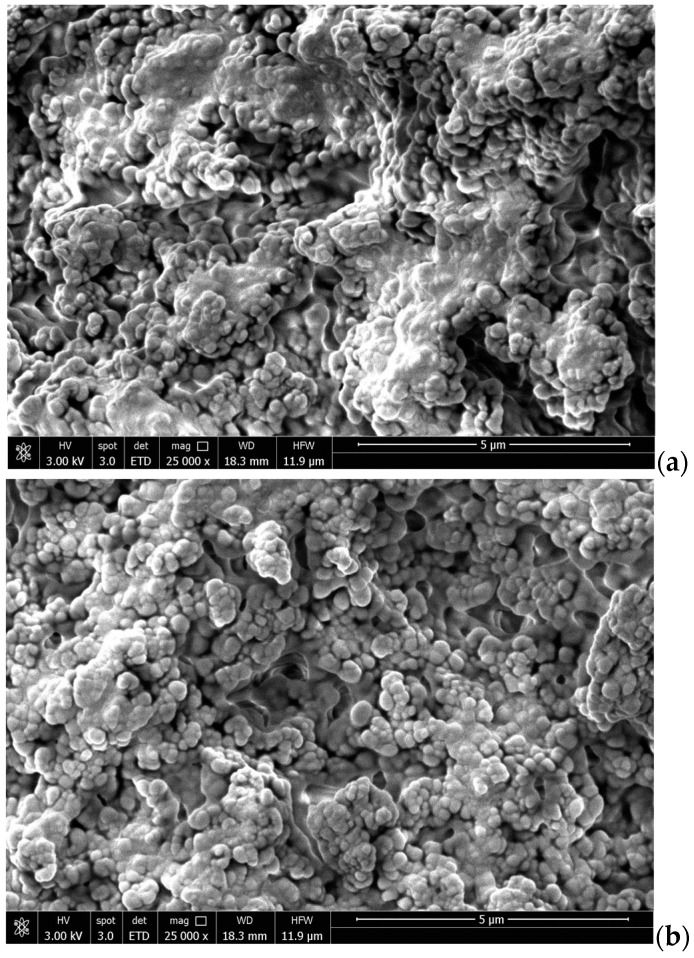
SEM images of PLA NPs loaded with α-TOC dried using (**a**) oven and (**b**) freeze-dryer.

**Figure 3 foods-13-00475-f003:**
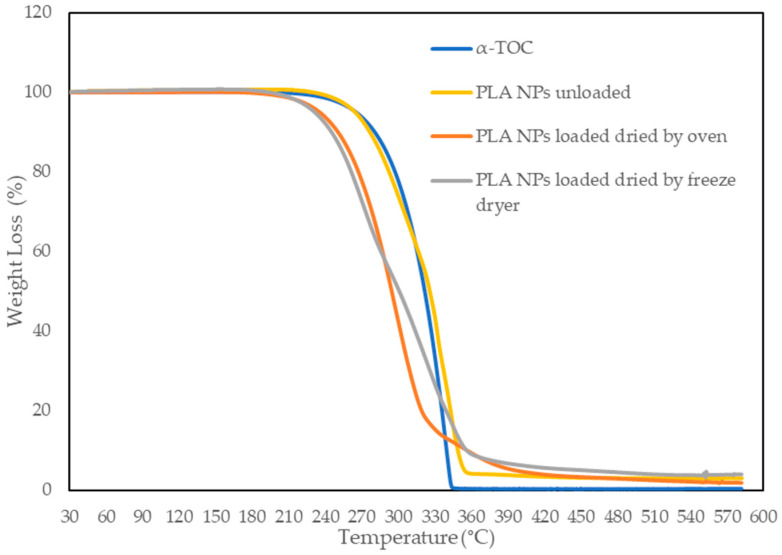
TG curves of α-TOC in free state, unloaded PLA NPs, and PLA NPs loaded with α-TOC dried using oven and freeze-dryer.

**Figure 4 foods-13-00475-f004:**
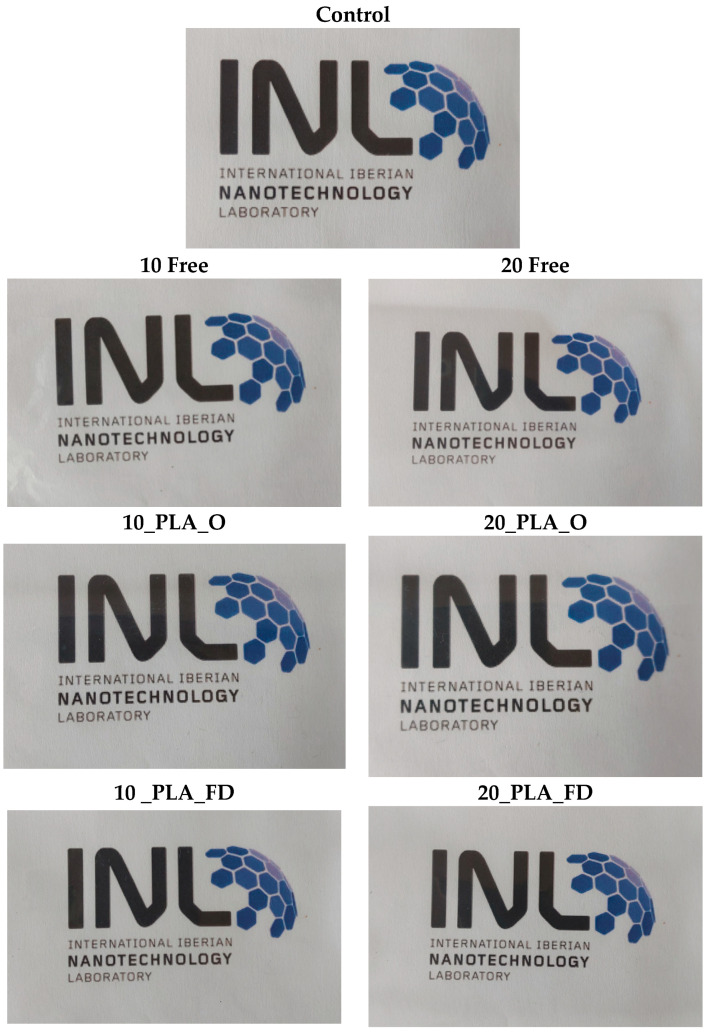
Pictures of films with a logo as background. LDPE film without (control) and with free-state α-TOC (10_Free and 20_Free) and loaded PLA NPs dried using oven (10_PLA_O and 20_PLA_O) and freeze-dryer (10_PLA_FD and 20_PLA_FD).

**Figure 5 foods-13-00475-f005:**
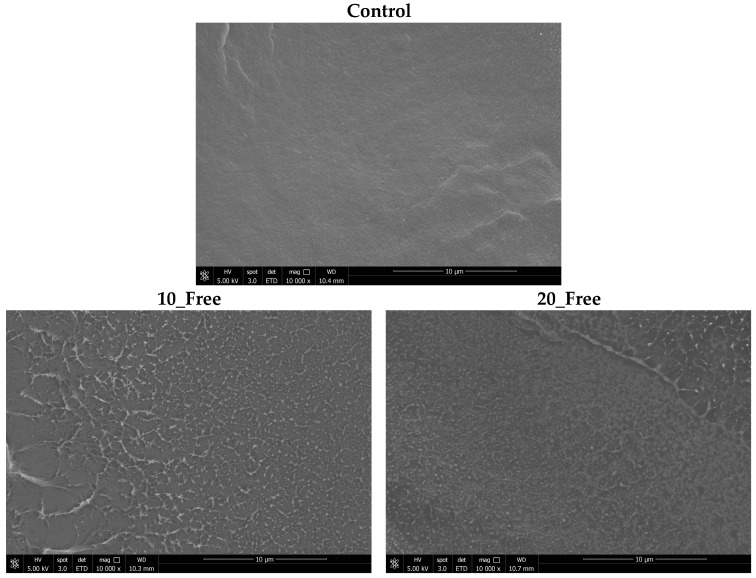
SEM images of LDPE films in cross-section without and with free-state α-TOC and loaded PLA NPs dried using oven and freeze-dryer.

**Figure 6 foods-13-00475-f006:**
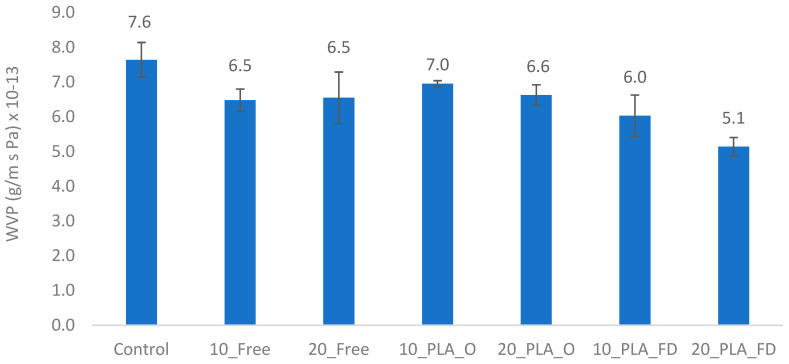
Water vapor permeability (WVP) of LDPE films without and with free-sate α-TOC and loaded PLA NPs dried using oven and freeze-dryer.

**Table 1 foods-13-00475-t001:** Film’ description and notation.

Film Description	Film Notation
LDPE film	Control
LDPE + 10 g/kg α-TOC free state	10_Free
LDPE + 20 g/kg α-TOC free state	20_Free
LDPE + 10 g/kg α-TOC NPs PLA (Oven)	10_PLA_O
LDPE + 20 g/kg α-TOC NPs PLA (Oven)	20_PLA_O
LDPE + 10 g/kg α-TOC NPs PLA (Freeze Dryer)	10_PLA_FD
LDPE + 20 g/kg α-TOC NPs PLA (Freeze Dryer)	20_PLA_FD

**Table 2 foods-13-00475-t002:** Average particle size, polydispersity index (PDI), and production yield for unloaded and loaded PLA nanoparticles (NPs).

Samples	Average Particle Size (nm)	PDI	Production Yield (%)
Unloaded PLA NPs	122.0 ± 4.4 a	0.123 ± 0.028 a	74.24 ± 9.76 a
Loaded PLA NPs	186.8 ± 9.6 b	0.149 ± 0.025 a	80.44 ± 9.10 a

Different letters in the same column indicate a statistically significant difference (*p* < 0.05).

**Table 3 foods-13-00475-t003:** Turbidity (%) and total color change (ΔE) for LDPE films without and with free-state α-TOC and loaded PLA NPs dried using oven and freeze-dryer.

Samples	Turbidity (%)	ΔE
Control	10.79 ± 0.30 a	-
10_Free	10.87 ± 0.50 a	0.37 ± 0.24 a,b
20_Free	12.03 ± 0.60 b	0.14 ± 0.03 a
10_PLA_O	12.80 ± 0.40 b	0.42 ± 0.10 a,b
20_PLA_O	15.13 ± 0.30 c	0.73 ± 0.21 b
10_PLA_FD	12.75 ± 0.20 b	0.26 ± 0.11 a
20_PLA_FD	14.74 ± 0.70 c	0.33 ± 0.11 a,b

Different letters in the same column indicate a statistically significant difference (*p* < 0.05).

**Table 4 foods-13-00475-t004:** Tensile strength (TS), Young’s Modulus (YM), and elongation at break (EB) measured in longitudinal direction (LD) and transversal direction (TD) of the LDPE films with free-state α-TOC and loaded PLA NP film.

Properties		Control	10_Free	20_Free	10_PLA_O	20_PLA_O	10_PLA_FD	20_PLA_FD
TS (MPa)	LD	17.5 ± 1.1 a	7.9 ± 0.3 b	8.2 ± 0.6 b,c	7.9 ± 0.3 b	8.7 ± 0.2 c	8.2 ± 0.3 b,c	8.6 ± 0.5 b,c
	TD	6.3 ± 0.6 a	8.7 ± 0.4 b	8.5 ± 0.8 b	8.5 ± 0.4 b	8.1 ± 0.2 b	8.6 ± 0.1 b	8.1 ± 0.2 b
YM (MPa) *	LD	356 ± 20 a	279 ± 33 b	284 ± 79 b	279 ± 33 b	264 ± 57 b	266 ± 72 b	189 ± 47 c
	TD	324 ± 28 a	227 ± 99 b	352 ± 4 a	348 ± 37 a	207 ± 64 b	236 ± 73 b	207 ± 64 b
EB (%)	LD	178 ± 33 a	259 ± 46 b	248 ± 69 a	259 ± 46 b	202 ± 51 a	208 ± 71 a	214 ± 70 a
	TD	67 ± 29 a	163 ± 74 b	217 ± 122 b	101 ± 47 a,b	57 ± 21 a	401 ± 3 c	57.6 ± 21 a

* Determined using secant 1%. Different letters in the same row indicate a statistically significant difference (*p* < 0.05).

**Table 5 foods-13-00475-t005:** Antioxidant activity coefficient (AAC) and total phenolic compounds (TPC) in LDPE films with free-state α-TOC and loaded PLA NPs films, after 3 and 10 days at 40 °C in contact with ethanol 95% (*v*/*v*).

	β-Carotene Assay (AAC)		TPC (GAE µg/mL)	
Samples	3 Days	10 Days	3 Days	10 Days
10_Free	243.25 ± 39.27 a	382.72 ± 7.09 ab	3.90 ± 0.62 a,c	2.83 ± 0.37 a
10_PLA_O	333.36 ± 24.10 b	320.12 ± 13.04 a	1.87 ± 0.28 c	7.17 ± 1.77 ab
10_PLA_FD	279.33 ± 15.46 a,b	299.22 ± 20.96 a	2.63 ± 0.45 a,c	7.86 ± 1.13 ab
20_Free	252.33 ± 34.46 a	443.96 ± 34.31 b	8.33 ± 0.67 b	6.93 ± 0.36 ab
20_PLA_O	295.29 ± 3.62 a,b	413.70 ± 42.43 ab	5.91 ± 1.4 a	7.55 ± 3.97 ab
20_PLA_FD	288.50 ± 34.60 a,b	341.40 ± 47.15 a	4.47 ± 0.49 a	11.65 ± 0.96 b

Different letters in the same column indicate a statistically significant difference (*p* < 0.05).

## Data Availability

The data presented in this study are available on request from the corresponding author.

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
