# Peer review of "Active Low-Density Polyethylene-Based Films by Incorporating α-Tocopherol in the Free State and Loaded in PLA Nanoparticles: A Comparative Study"

_foods, 2024, doi:10.3390/foods13030475_

Round 1

Reviewer 1 Report

Comments and Suggestions for Authors

manuscript foods-2775088 for Foods

Dear editor,

The authors in the manuscript report (Active low-density polyethylene-based films by incorporating α-tocopherol in the free state and loaded in PLA nanoparticles: a comparative study). The paper needs improvement and can be accepted after taking care of the following comments:

1. The abstract was not exciting. The authors must be writing in a short with the main essential results

2. For sure, this introduction can be accepted in this form. It is too short like a letter. So, the author must be divided the introduction to discuss the following items: problem, the type of material, the used technique, the literature survey, the point of view, and the novelty. So, the authors ((must)) be using these papers and similar than these work to be support and cited to increase the value of this work:

https://doi.org/10.1016/j.foodchem.2023.138094

https://doi.org/10.1111/jfpe.13088

3. The aim of the work should be highlighted in the introduction part. The author just mentioned the characterization of the alpha-tocopherol (α-TOC) was added to low-density polyethylene (PE-LD) using two strategies. α-TOC was incorporated in the free form and loaded in poly(lactic acid)-based nanoparticles. Why do we need to two approaches?

4. In the experimental part, what is the dimension of the PLA nanoparticles.

5. In the results of mechanical properties? Need stress-strain cure for explain the tensile, young modulus and elongation of the materials.

6. How PLA nanoparticles were obtained to be employed as antioxidant activities of PE-LD films.

7. In Figure 5 (a) (b) apparently, the scale (10 µm) is not the same with (b, c). This image needs to be reviewed. The PLA was used as a nanoparticle in this work, which explains why a micro size was shown in the SEM image.

8. The SEM images are blurry. It is better to use good-quality images. And explain physically why do they get a difference in the particle size.

9. What is the role of PE-LD in antioxidant activity 

10. The conclusion should be revisited to highlight the main findings of the paper.

11. As a whole the manuscript needs major revision.

Comments on the Quality of English Language

ok

Reviewer 2 Report

Comments and Suggestions for Authors

The manuscript presents the results of work on the modification of PELD films. The modification was carried out by direct introduction of tocopherol or tocopherol encapsulated in PLA nanoparticles.
The manuscript has been reliably prepared, the proposed research work is suitable for the assumptions of using the material as food packaging Nevertheless, I have several questions and doubts.
 Please check the calculation of the amount of tocopherol in the PE matrix on page 5 (line 191), according to me it should be 0.02 g of active agent per g of material.
Why was such a significant amount of compatibiliser used in relation to the modifier used?
Why is the stability of the PLA nanoparticles with tocopherol lower than that of the individual components?
Were the PLA nanoparticles with tocopherol not destroyed/melted during the manufacture of the PE film? On SEM images their structures in PE are not visible.
How was the turbidity determined?
Error in text line 425.
